# Metabolomics-Based Profiling via a Chemometric Approach to Investigate the Antidiabetic Property of Different Parts and Origins of *Pistacia lentiscus* L.

**DOI:** 10.3390/metabo13020275

**Published:** 2023-02-14

**Authors:** Chabha Sehaki, Roland Molinie, David Mathiron, Jean-Xavier Fontaine, Nathalie Jullian, Fadila Ayati, Farida Fernane, Eric Gontier

**Affiliations:** 1BIOPI-UPJV Laboratory UMRT BioEcoAgro INRAE1158, SFR Condorcet FR CNRS 3417, UFR of Sciences, University of Picardie Jules Verne, 33 Rue Saint Leu, 80000 Amiens, France; 2Laboratory of Natural Resources, University Mouloud Mammeri of Tizi-Ouzou, Tizi-Ouzou 15000, Algeria; 3Analytical Platform, UFR of Sciences, University of Picardie Jules Verne, 33 Rue Saint Leu, 80000 Amiens, France

**Keywords:** *P. lentiscus*, α-glucosidase, phytochemical profiling, UHPLC-ESI-HRMS, metabolomic approach, antidiabetic

## Abstract

*Pistacia lentiscus* L. is a medicinal plant that grows spontaneously throughout the Mediterranean basin and is traditionally used to treat diseases, including diabetes. The aim of this work consists of the evaluation of the α-glucosidase inhibitory effect (i.e., antidiabetic activity in vitro) of different extracts from the leaves, stem barks and fruits of *P. lentiscus* harvested on mountains and the littoral of Tizi-Ouzou in Algeria. Metabolomic profiling combined with a chemometric approach highlighted the variation of the antidiabetic properties of *P. lentiscus* according to the plant’s part and origin. A multiblock OPLS analysis showed that the metabolites most involved in α-glucosidase inhibition activity were mainly found in the stem bark extracts. The highest inhibitory activity was found for the stem bark extracts, with averaged inhibition percentage values of 84.7% and 69.9% for the harvested samples from the littoral and mountain, respectively. On the other hand, the fruit extracts showed a lower effect (13.6%) at both locations. The UHPLC-ESI-HRMS characterization of the metabolites most likely responsible for the α-glucosidase-inhibitory activity allowed the identification of six compounds: epigallocatechin(4a>8)epigallocatechin (two isomers), (epi)gallocatechin-3′-O-galloyl-(epi)gallocatechin (two isomers), 3,5-O-digalloylquinic acid and dihydroxy benzoic acid pentoside.

## 1. Introduction

Statistics on people diagnosed with diabetes reveal that diabetes is a worldwide disease [1]. This disease affects approximately 5% of the world’s population [2]. Diabetes mellitus (type 1) is characterized by chronic hyperglycemia caused by insufficient insulin secretion [3], whereas diabetes (type 2) is a metabolic disease resulting from the dynamic interaction between defects in insulin secretion and insulin action [2]. This insulin deficiency due to the dysfunction of pancreatic β cells [4] leads to increased glucose concentrations in the blood. Several drugs, such as sulfonylureas and biguanides, are used to lower blood glucose levels in diabetics. Among these classes, there are inhibitors of α-glucosidase, an enzyme responsible for the hydrolysis of carbohydrates to allow the passage of glucose into the bloodstream [5]. Currently, acarbose, miglitol and voglibose belong to classes of competitive α-glucosidase inhibitors in the intestine [6], reducing postprandial hyperglycemia. However, the use of these drugs has shown many side effects and toxicities [7]. It causes diarrhea [8] and adverse digestive disorders [8,9]. It has also been reported that acarbose can induce hepatitis [10,11] and increase liver enzyme levels [12].

Recent scientific research has shown that natural products could have an important hypoglycemic effect and could be alternatives for the treatment of type 2 diabetes. The plant families, which enclose the species, studied the most as inhibitors of α-glucosidase, based on a review by Benalla et al. (2010) [1], are Fabaceae, Crassulaceae, Hippocrateacae, Lamiacea and Myrtaceae. These studies have revealed the importance of medicinal plants as natural sources of antidiabetic bioactive agents. *P. lentiscus* (lentisk or mastic tree), a medicinal plant growing in the wild in the Mediterranean area [13,14] and belonging to the Anacardiaceae family, is traditionally used to treat hypertension, cough, sore throat, eczema, stomach ache, toothache, kidney disorders, jaundice [15], skin problems and diarrhea [16,17], but also diabetes [18]. In the ethnobotanical investigation on antidiabetic plants conducted in Morocco by Bouyahya et al. (2021), it was revealed that lentisk leaves were the most used part. In traditional medicine, for the treatment of diabetes, the administration of lentisk extract is conducted orally; leaf decoction is the main mode of preparation, followed by infusion. The phytochemistry of lentisk has been extensively studied using high-performance liquid chromatography–mass spectrometry (HPLC-MS). Several phenolic acids have been detected, such as gallic acid, quinic acid and galloyl derivatives [19,20,21,22,23], as well as flavonoid glycoside compounds [21].

Recent investigations have focused on the study of phytochemical profiling and the multibiological potential of plant extracts via integrated approaches [24,25]. The researchers in these studies were interested in metabolomic analyses to characterize metabolites in plants that may be responsible for particular bioactive activities, such as antioxidant and in vitro antidiabetic activities [26,27,28].

Even though there are many ethnobotanical pharmaceutical uses of *P. lentiscus* [29], very few studies [23,30] have been carried out to explore the possibilities of using this species as an antidiabetic agent. These studies were carried out by Foddai et al. [30] on aqueous leaf and fruits extract and Mehenni et al. [23] on ethanolic extracts. Most of the research on P. lentiscus has been conducted to find out its antioxidant [31,32,33,34,35,36,37], antimicrobial [38,39,40,41], anticancer [22,42,43,44] and anti- inflammatory potential [19,45,46,47]. So far, no study has reported a metabolomic analysis to identify inhibitors of antidiabetic potential in different lentisk organs. It is known that different geographical regions can affect the composition and biological activity of plants [48]. Therefore, our study is the first work that aimed to achieve the following:(i)Evaluate α-glucosidase inhibition in *P. lentiscus* samples and identify compounds more positively correlated to α-glucosidase inhibition using a metabolomic approach;(ii)Study the variation of this inhibition according to the organ (i.e., leaf, stem bark and fruit) and the geographical origin (i.e., mountain and littoral) of the plant *P. lentiscus.* This metabolomic method was based on the link between phytochemical profiling by ultra-high-performance liquid chromatography coupled to high-resolution mass spectrometry (UHPLC-ESI-HRMS) and α-glucosidase dosage data.

This study could contribute to the valorization of the antidiabetic potential of this resource in the pharmaceutical field through, for example, the elaboration of traditionally improved drugs (TID) in developing countries. The antidiabetic potential of this plant could also be exploited in the field of food supplements.

## 2. Materials and Methods

### 2.1. Chemicals

The main chemicals used for the measurements of α-glucosidase inhibition were α-glucosidase from *Bacillus stearothermophilus* (G3651-250 UN, 84 units/mg, 86% protein), p-nitrophenyl α-D-glucopyranoside (N1377, ≥99.0%) and acarbose (PHR1253-Pharmaceutical Secondary Standard; Certified Reference Material). The analytical standards for the UHPLC-ESI-HRMS analysis were (−)-epigallocatechin (EGC) (≥90%, 08108-supeclo), myricitrin (≥99.0%, 91255, Sigma-Aldrich, Saint Quentin Fallavier, France),apigenin 7-glucoside (≥97.0%, 44692, Supelco) and tiliroside (≥95.0%, PHL89809-phyproof^®^Reference Substance). These analytical-grade chemicals used were purchased from Supelco and Sigma-Aldrich, Saint Quentin Fallavier, France.

### 2.2. Plant Collection

The plant species *P. lentiscus* was identified using the flora of Quézel and Santa (1962–1963). In addition, the lentisk samples were deposited in the herbarium of the National Superior School of Agronomy of Algeria. This identification was confirmed by Benhouhou and resulted in the issuance of a certificate attesting that this plant was indeed *P. lentiscus.* The samples of leaves, stems and fruits of *P. lentiscus* were collected separately from plants in the Algerian region of Tizi-Ouzou. Its relatively contrasted climatic regimes are representative of those of the large area of northern Algeria between the littoral and mountains. For this study, we selected two geographical sites in the Tizi-Ouzou region: Ait-Irane on the mountain and Tigzirt on the littoral. In all, 54 samples were collected. The information of the 54 samples, such as the collected site, collected date, GPS of each collection site and morphological variation in the characters of each sample, is provided in Table 1. The lentisk samples harvested from the two sites (mountain and littoral) do not show remarkable morphological differences according to the difference in altitude of the harvesting sites (Table 1). 

The Mediterranean climate in Ait-Irane is characterized by a dry summer and relatively low temperatures during winter (Appendix A). On the other hand, the summer and winter temperatures are less extreme, and the humidity is higher throughout the year in Tigzirt (Appendix A). At each site, a set of nine individual female trees was selected for all of the different samples. The collection was conducted at sunrise in autumn (6–8 a.m.) in October 2019.

### 2.3. Drying and Extraction

The samples from each organ were cleared of debris and air-dried in the dark for 7 days at 25–30 °C. The samples were ground using a ball mill. Fifty-four powdered samples were obtained (9arbusts*2sites*3organs). A methanol extraction of each powder (200 mg) from each organ was performed under heating at 70 °C for 30 min in an Eppendorf Thermo Mixer. The resulting mixture was centrifuged at 1100× *g* for 5 min. The recovered supernatant was evaporated under nitrogen. The dry crude extract recovered after the evaporation of methanol was dissolved in a 50:50 (*v/v*) water/cyclohexane mixture. After centrifugation, the aqueous phase was collected and lyophilized [49]. The lyophilized aqueous extract was stored at −20 °C for LC-MS analysis and biological activity evaluation.

### 2.4. UHPLC-ESI-HRMS Analysis of the Extracts

Each lyophilized aqueous extract (54 extracts) was dissolved in 2 mL of a 50:50 (*v/v*) water/methanol mixture. After shaking and sonication for one minute, the solutions were diluted to 1:10 in a 50:50 (*v/v*) water/methanol mixture for UHPLC-ESI-HRMS analysis. The UHPLC-ESI-HRMS analysis conditions were set according to Tchoumtchoua et al. (2019) [50], with some slight modifications.

The UHPLC-ESI-HRMS analyses were performed on an ACQUITY UPLC I-Class system coupled to a high-resolution hybrid Vion IMS QTOF (ion mobility quadrupole time-of-flight) mass spectrometer equipped with an electrospray ionization (ESI) source (Waters, Manchester, UK). One microliter of each sample was injected, and the chromatographic separation was performed on a Kinetex Biphenyl column (100 × 2.1 mm, 1.7 µm) (Phenomenex, Torrance, CA, USA) maintained at 55 °C. The mobile phase flow rate was set at 0.5 mL/min, and a gradient elution from water with 0.1% formic acid (A) to methanol with 0.1% formic acid (B) was programmed as follows (A:B): 80:20 (t = 0 min), 80:20 (t = 0.5 min), 40:60 (t = 5 min), 10:90 (t = 6 min), 10:90 (t = 7 min), 80:20 (t = 7.5 min) and 80:20 (t = 10 min). The ESI source was set to a capillary voltage of 2.4 kV in the negative and positive ionization modes, with a sample cone voltage of 20 V. The source and desolvation temperatures were set to 120 and 450 °C, respectively. Nitrogen was used as the desolvation and cone gas at flow rates of 1000 and 50 L/h, respectively. For accurate mass measurements, lockmass correction was applied using the [M-H]^−^ ion at an *m*/*z* of 554.2615 from a solution of leu-enkephalin (100 pg/L in H_2_O/CH_3_CN (50:50, *v/v*) with 0.1% formic acid. The TOF was operated in the sensitivity mode, providing an average resolving power of 50,000 (FWHM). The HDMS^E^ spectra were recorded in the profile mode over amass range of 50 to 2000 *m*/*z*, with the scan time set to 0.2 s. HDMS^E^ acquisition works by alternating two functions, a so-called LE (low-energy) one with a collision energy set at 6 eV to obtain non-fragmented spectra with information on precursor ions and a HE (high-energy) one with a ramp of collision energy values from 25 to 50 eV to obtain fragmented ion spectra thanks to an alignment by ion mobility of the fragments that had the same drift time as the precursors they came from. The analytical standards for the UHPLC-ESI-HRMS analysis were injected.

### 2.5. Glucosidase-Inhibitory Activity

The measurement of effect of the extract on the catalytic activity of α-glucosidase was performed according to the method of Bachhawat et al. (2011) [51], with a slight modification. In a 96-well plate, a reaction mixture containing 50 μL of phosphate buffer (50 mM, pH = 6.8), 10 μL of α-glucosidase (1 U/mL) (α-glucosidase from *Bacillus stearothermophilus*) and 20 μL of varying concentrations of extract (0, 2, 4, 6, 8 and 10 µg/mL) was preincubated at 37 °C for 15 min. Then, 20 μL of p-nitrophenyl α-D-glucopyranoside (PNPG) (1 mM) was added as a substrate, and the mixture was incubated again at 37 °C for 20 min. The reaction was stopped by adding 50 μL of sodium carbonate (0.1 M). The yellow color produced was read at 405 nm using a plate reader. Acarbose at various concentrations (0.2–1 mg/mL) was included as a positive control. A negative control without extracts was assayed in parallel. The experiment was performed three times for each sample. The UHPLC-ESI-HRMS injected standards were also tested for their effect on α-glucosidase inhibition. The results are expressed as percentages of inhibition, which were calculated according to the following formula (with A corresponding to the absorbance):$$\mathrm{Inhibition}\left(\%\right)=\frac{\left(A_{\;\mathrm{negative}\;\mathrm{control}}-A_{\;\mathrm{sample}}\right)}{{A\;}_{\mathrm{negative}\;\mathrm{control}}}\times 100$$
where A _negative control_ corresponds to the absorbance of the control mixture (mixture with the buffer instead of the inhibitor), and A _sample_ represents the absorbance of the samples containing an inhibitor (extracts or acarbose). The α-glucosidase inhibitions of the standards injected for the UHPLC-ESI-HRMS analysis were also tested.

### 2.6. Metabolomic and Chemometric Analysis

Data processing was performed with the UNIFI software V1.9.4 (Waters, Manchester, UK). The multivariate matrix consisted of the ions detected in the samples after the UHPLC-QTOF-MS analysis in the negative (neg) and positive (pos) ionization modes (see Appendix A) in the retention time range (0.5–5 min). Only the ions with a response greater than 2000 (peak area) were retained in at least one condition (i.e., leaves, stem bark and fruit). Among these ions, those correlated with each other (r ≥ 0.80) at the same retention time were removed. Consensus orthogonal partial least squares (multiblock OPLS) was used as the chemometric tool to model the information in the rich chemical and biological data elaborated in this study [52]. This multiblock OPLS analysis was performed using the latest version of MATLAB 2020b. The UHPLC-QTOF-MS chromatograms in the negative and positive ion modes and the percentage of α-glucosidase inhibition were used for untargeted analysis. The multiblock OPLS showed the relationship between the responses (I.e., peak area) of the metabolites detected via the UHPLC-QTOF-MS chromatograms of the extracts analyzed and the intensity of the observed α-glucosidase inhibitory effect. 

### 2.7. Intensity Variation of the Metabolites Identified

The objective of the univariate analysis was to study the variation of (i) the inhibition of the α-glucosidase enzyme and (ii) the intensity of the metabolites most likely responsible for this inhibition according to the part of the lentisk used and its geographical origin. These statistical analyses were performed by pairwise comparison through a post hoc Kruskal–Wallis test adjusted with Bonferroni correction using the package PMCMRplus version1.9.4 [53] for analysis of the variances and the package MulticompView version 0.1–8 for analysis of the groups [54]. Significant differences were considered at the 95% confidence level. These analyses were performed with the R software version R (≥3.5.0).

## 3. Results

In the present work, the different aerial parts (i.e., leaves, stem barks and fruits) of the plant *P. lentiscus* were collected from two sites (i.e., mountain and littoral) in Tizi-Ouzou (Algeria). The 54 samples obtained were characterized phytochemically and biologically. Firstly, the evaluation of the α-glucosidase inhibition property in vitro of all of the extracts was performed. Secondly, metabolite profiling was performed using an up-to-date analytical technique: UHPLC-ESI-HRMS. Finally, a multivariate metabolomic analysis (multiblock OPLS) was performed in the following order:

(i) Study of the α-glucosidase inhibition differences between the aerial parts and between the geographical origins of *P. lentiscus*;

(ii) Characterization of the metabolites most involved in the inhibition of α-glucosidase.

### 3.1. α-Glucosidase-Inhibitory Activity

The inhibitory effect on α-glucosidase was estimated by evaluating the percentage of inhibition of this enzyme in a 96-well plate. Figure 1 presents the percentages of inhibition of α-glucosidase by the different lentisk extracts (i.e., leaves, stem barks and fruits at a concentration of 10 μg/mL) harvested from the two locations in Tizi-Ouzou (i.e., littoral and mountain). The stem bark extracts showed stronger inhibitory activity, and the average values were 84.7 ± 5.9% (IC50 = 5.8 ± 0.4) and 69.9 ± 19.9% (IC50 = 7.9 ± 3.3) at the littoral and mountain, respectively. The fruit extracts showed a lower effect (13.6 ± 6.2%) at both locations. The pairwise comparison through the Kruskal–Wallis test adjusted with Bonferroni correction [53,54] allowed for the study of the significance of the inhibition variation according to the organ and also the location. Depending on the location of the plant, the variation of the inhibitory effect was not significant for each type of organ. On the other hand, depending on the part of the plant used, the inhibition difference was significant between the stem barks and fruits and between the leaves and fruits.

We also tested acarbose (reference control; α-glucosidase inhibitor), a drug used for the treatment of type 2 diabetes. The inhibition by acarbose of α-glucosidase was approximately 100% (see Figure 1). Comparing the effect of the lentisk extracts with acarbose, we noticed that the inhibitory effects of acarbose and stem bark extracts on α-glucosidase were comparable. The difference between the two effects was not significant (*p*-value > 0.05) according to the Kruskal–Wallis test following pairwise comparisons using Dunn’s many-to-one test [53]. This same test revealed that the difference in α-glucosidase inhibition was not significant between acarbose and leaf extracts, but it was significant between acarbose and fruit extracts (*p*-value < 0.05). A figure showing the same results by an IC50 calculation is included in the Appendix A.

### 3.2. Metabolomic and Chemometric Analysis 

In order to objectively assess the differences in the α-glucosidase inhibition and chemical profile depending on the plant part and its geographical origin, the supervised approach, consensus orthogonal partial least squares (multiblock OPLS) based on UHPLC-ESI-HRMS profiling and in vitro enzyme inhibitory bioactivities, was performed. This chemometric approach also aimed to characterize the metabolites positively correlated with α-glucosidase inhibition.

The UHPLC-QTOF-MS data of 54 extracts were analyzed, and the values of the α-glucosidase-inhibitory activities were processed using multivariate data analysis (multiblock OPLS). The final matrix that was used for the multiblock OPLS analysis was composed of 198 ions in the negative ionization mode and 160 ions in the positive one. The quality parameters of the multiblock OPLS model (R^2^ = 0.74 and Q^2^ = 0.72) demonstrate the good fit of this model. The two ion blocks (i.e., negative ionization mode (neg) and positive ionization mode (pos)) contributed in the same way to the constitution of the multiblock OPLS model (see Appendix A). The multiblock OPLS analysis generated two graphs: a score plot (Figure 2A) and a loading plot (Figure 2B). The two components (i.e., predictive and orthogonal) of the score plot showed a total variance of 81.5%, with 75% for the predictive component and 6.5% for the orthogonal component. This score plot separated the 54 samples (i.e., leaves, stem barks and fruits) according to the part of the plant used and the sampling location (i.e., littoral or mountain) into six different groups. The loading plot shows the metabolites detected in these extracts. The inhibitory effect increases going to the right of the predictive component of the loading plot. Therefore, we were interested in identifying metabolites located on the right end of the loading plot, with coordinates greater than five on the predictive component (see the values of these coordinates in Appendix A). These metabolites were the ones most involved in the inhibition of α-glucosidase in the different lentisk extracts (Figure 2B). These metabolites were present in intense concentrations in the stem barks (Figure 2A,B).

The tentative Identification of the metabolites most positively correlated with α-glucosidase inhibition was based on UHPLC-QTOF-MS (RT, *m/z*, molecular formula) data. Using the characteristic fragments obtained after the HDMS^E^ analysis, we were able to propose fragmentation patterns identifying each molecule. The identification process was also based on comparisons using on the available databases, including the KnaPSacK metabolome database, MassBank, Kyoto Encyclopedia of Genes and Genomes (KEGG) and MetFrag, as well as the UNIFI library. This identification was also confirmed using previous publications on the same molecules. Following this identification methodology, we identified four metabolites, shown in color in Figure 2B, in the negative and positive ionization modes and two other metabolites in the negative ionization mode.

### 3.3. UHPLC-ESI-HRMS Identification of α-Glucosidase Inhibitory Metabolites 

The *P. lentiscus* extracts were found to have an α-glucosidase inhibition effect that was more or less impactful according to the plant’s living environment (i.e., mountain or littoral) and the considered organ (i.e., leaves, stem barks or fruits). With the aim of obtaining a better understanding of this effect, UHPLC-ESI-HRMS analyses were performed to characterize the phytochemical content that could be responsible for α-glucosidase inhibition. Statistical analyses (i.e., multiblock OPLS) were used for comparing the data of the different extracts and to highlight those for which the metabolic content could be correlated with strong α-glucosidase inhibition activity. This approach led to a total of six compounds. These compounds were putatively identified as phenolic compounds by ESI-HRMS in the HDMS^E^ mode based on accurate mass measurements of the corresponding ions and their fragments. The UHPLC-ESI-HRMS data results are summarized in Table 2 and Appendix A. Indeed, these tables include the retention time (RT), observed *m/z*, mass error value (ppm), molecular formula, characteristic fragment ions used for identification with their intensities and the suggested identity of each compound in agreement with previous studies. Appendix A contains more information than Table 2.

According to these HDMS^E^ data, the six different metabolites have been widely identified as:

Epigallocatechin(4a->8)epigallocatechin, or a dimer of (epi)gallocatechin (2×EGC): The HDMS^E^ spectrum of metabolite 1 (RT 0.45 min) revealed a deprotonated pseudomolecular ion [M-H]^−^ at *m/z* 609.1242, allowing us to access the molecular formula C_30_H_26_O_14_. All of the fragment ions obtained were found to be typical of the proanthocyanidin family. Heterocyclic ring fission (HRF), retro-Diels–Alder (RDA) and quinone-methide (QM) were the characteristic fragmentation mechanisms observed for this class of molecule. Indeed, RDA and HRF fragmentation provide information on bond hydroxylation following ring cleavage, while interflavan bond cleavage, also called quinone methide (QM) fragmentation, can reveal information on the nature of monomeric units. A proposed fragmentation pathway describing the formation of the characteristic fragment ions for this molecule is depicted in Figure 3. First, the fragmentation of the [M-H]^−^ion at *m/z* 609.1242 occurred after a neutral loss of 168 Da to the low-intensity fragment ion at *m/z* 441.0825 by the RDA mechanism in the C ring. Thereafter, the loss of a water molecule (−18 Da) from the ion at *m/z* 441.0825 was observed to result in the ion at *m/z* 423.0722, which was the base peak in the spectrum. The fragment ion at *m/z* 305.0667 was obtained from *m/z* 609.1242 by an interflavan bond cleavage (QM) between the C and D rings, evidencing the nature of this compound as one composed of two units having the same mass, supporting the hypothesis of a dimer of (epi)gallocatechin [56,61]. The presence of the fragment ion at *m/z* 125.0241 can be explained by the HRF mechanism from the corresponding ion of (epi)gallocatechin at *m/z* 305 releasing phloroglucinol from the A ring [62]. The other fragments detected at *m/z* 177, 219, 261 and 285 were characteristic ions produced from the fragmentation of *m/z* 305 [43,62]. It should be noted that epigallocatechine dimers composed of two (epi)gallocatechin units have been previously identified in grape canes and red wine [56,61,63] but never in *P. lentiscus* extracts so far.

Epigallocatechin(4a->8)epigallocatechin (isomer of metabolite 1): Metabolite 4 (RT 0.61 min) was putatively identified by HDMS^E^ as an isomer of metabolite 1 (epigallocatechin(4a->8)epigallocatechin, a dimer of epigallocatechine), as the same molecular formula, C_30_H_26_O_14_, was found from the corresponding [M-H]^-^ ion at *m/z* 609.1233. Moreover, the same characteristic fragment at *m/z* 423.0716 was obtained from the precursor ion at *m/z* 609.1233 by the loss of an RDA fragment (−168 Da) followed by the loss of water (−18 Da) (Figure 3).

(Epi)gallocatechin-3′-O-galloyl-(epi)gallocatechin: Metabolite 3 (RT 0.56 min), detected at *m/z* 761.1339 with the corresponding molecular formula C_37_H_30_O_18_, was identified as (epi)gallocatechin-3′-O-galloyl-(epi)gallocatechin. Firstly, the fragmentation of the [M-H]^-^ion at *m/z* 761.1339 resulted in the fragment at *m/z* 609.1228 by the loss of 152 Da corresponding to a galloyl residue. For this last fragment ion, the molecular formula C_30_H_26_O_14_ was obtained, revealing that this metabolite could have an epigallocatechin dimer structure type. Indeed, the typical fragment ions at *m/z* 423.0820 and 305.0568, as previously described for epigallocatechin dimers, were observed. In addition, two other fragments were observed on the fragmented HDMS^E^ spectrum in agreement with the structure proposed for metabolite 3, namely, the ion at *m/z* 591.1134, which was produced by the loss of a water molecule (−18Da) from the *m/z* 609.1228, and the ion at *m/z* 465.0820 obtained by the loss of a phloroglucinol (−126 Da) molecule at *m/z* 591.1134. The details of the fragmentation mechanism and of the characteristic fragmentation structures discussed above can be found in Figure 4a,b.

From the preceding points, this compound was identified as (epi)gallocatechin-(4,8′)-3′-O-galloyl-(epi)gallocatechin [58]. This compound has never been detected in *P. lentiscus* extracts before.

**(Epi)gallocatechin-3′-O-galloyl-(epi)gallocatechin (isomer of metabolite 3):** Metabolite 5 (RT 0.72 min) was putatively identified by HDMS^E^ as an isomer of metabolite 3 ((epi)gallocatechin-3′-O-galloyl-(epi)gallocatechin), as the same molecular formula, C_37_H_30_O_18_, was found from the corresponding [M-H]^−^ ion at *m/z* 761.1344. Moreover, the same characteristic fragment ion at *m/z* 423.0716 was obtained from the precursor ion at *m/z* 609.1233 by the loss of an RDA fragment (−168 Da) followed by the loss of water (−18 Da) (Figure 4b). All of these observations are in agreement with a structure that could be an isomer of (epi)gallocatechin-3′-O-galloyl-(epi)gallocatechin.

**3,5-O-digalloylquinic acid**: Metabolite 2 (RT 0.55 min) was putatively identified as 3,5-O-digalloylquinic acid. It was characterized by an [M-H]^−^ ion at *m/z* 495.0774, enabling us to find the molecular formula C_21_H_20_O_14_. Under fragmented HDMS^E^ conditions, the [M-H]^−^ ion at *m/z* 495.0774 underwent two successive losses of a galloyl moiety (−152 Da), leading to fragment ions at *m/z* 343.0661 and 191.0551. The accurate mass measurement for the fragment at *m/z* 191.0551, with the determination of its molecular formula as C_7_H_11_O_6_, confirmed the quinate structure, and for the fragment ion at *m/z* 169.0136 with the corresponding molecular formula C_7_H_5_O_5_, a gallate structure was determined [42,43,64]. A proposed fragmentation pathway is described in Figure 5.

This compound, labeled metabolite 2, was recently identified in *P. lentiscus* leaves [43,64].

**Dihydroxy benzoic acid pentoside:** Metabolite 6 (RT 0.73 min) was characterized by an [M-H]^−^ ion at *m/z* 285.0608, enabling us to find the molecular formula C_12_H_14_O_8_ for this compound. Under fragmented HDMS^E^ conditions, the [M-H]^-^ion at *m/z* 285.0608 first underwent a loss of dehydrated pentose (−132 Da) to provide hydroxybenzoate, which was immediately followed by the loss of a radical species, CO_2_H (−45 Da), leading to the radical anion at *m/z* 108.0209 [60]. All of the structures of the fragments and the losses are presented in Figure 6. Thus, metabolite 6 was putatively identified as dihydroxy benzoic acid pentoside.

### 3.4. Intensity Variation of the Metabolites Identified

The univariate analysis generated box plots of the metabolites identified (in negative ionization, Figure 7). Each box plot presents the contents of a metabolite in the different lentisk parts from the mountain and littoral. The metabolite intensities were expressed as the area of the chromatographic peak corresponding to each metabolite in the extracts. The different letters indicate that the intensities were significantly dissimilar after a post hoc Kruskal–Wallis test (*p*-value < 0.05) for each compound. This statistical test showed that the location of the plant had no significant effect on the variation of the intensity of the metabolites detected in the lentisk’s different parts.

According to the intensities of the identified metabolites (i.e., chromatographic peak areas), we can say that the samples from the littoral had higher concentrations of α-glucosidase-inhibitory compounds, especially the stem barks. These results are consistent with those of the multiblock OPLS analysis.

## 4. Discussion

In the present work, *P. lentiscus* was studied for its use as a hypoglycemic agent in traditional medicine. This is the first study that reports the phytochemical screening of α-glucosidase-inhibiting compounds. These compounds were present in the different parts of *P. lentiscus* harvested in two different regions. The work was based on a chemometric study that linked biologic activity to data from a UHPLC-ESI-HRMS analysis.

Our results show the variability of the inhibition of α-glucosidase in regards to the organ and the geographical origin of Algerian lentisk. The α-glucosidase inhibition of lentisk stem bark was measured in this study. Compared to the previous studies in the literature, the fruit extracts (IC50 = 42 µg/mL) showed a greater inhibitory potency than did Sardinian *P. lentiscus* fruit aqueous extracts, with IC50 = 230 µg/mL [30].The inhibition percentages of α-glucosidase of the leaves and fruit extracts (49% and 13%, respectively) were comparable to the inhibition percentages of α-amylase of ethanolic leaf (55%) and fruit (54%) extracts [23].The activity of α-glucosidase inhibition has been tested in several other plants [65,66,67]. Plant extracts including *Tamarix nilotica* (IC50 = 33.3 µg/mL) [65], *Tetracera scandens* (IC50 = 3.4 µg/mL) [66] and *Cosmos caudatus* (IC50 = 39.1 µg/mL) [67] have shown promising α-glucosidase inhibition. These activities are comparable to those of acarbose and to the results found in the present study. In our case, the geographical origin did not have a significant effect on the variation of the α-glucosidase inhibition. A previous study revealed that *Parkia speciosa* pods collected from different locations in Malaysia show different levels of α-glucosidase-inhibitory activity [68]. Our results indicate that the change in the α-glucosidase inhibition activity was more significant according to the part of the plant than its habitat. This is explained by the variation in the chemical compositions of the different extracts. Multivariate models can show the predictive power of metabolomic analysis for α-glucosidase inhibition and can indicate which groups of metabolites contribute most to this activity [24,25]. Therefore, the metabolomic approach applied in our study allowed us to characterize the metabolites that contributed more to the inhibition of α-glucosidase enzymatic activity. A total of six compounds were identified, of which four were flavonoids derived from epigallocatechin. To the best of our knowledge, epigallocatechin(4a->8)epigallocatechin (two isomers), (epi)gallocatechin-3′-O-galloyl-(epi)gallocatechin (two isomers) and dihydroxy benzoic acid pentoside, but not 3,5-O-digalloylquinic acid, were identified in the different extracts (i.e., leaves, stem barks and fruits) of *P. lentiscus.* The monomer epigallocatechin,the basic structure of four molecules among these metabolites, has been identified previously in lentisk leaves [19,64]. Epigallocatechin, analyzed by UHPLC-ESI-HRMS as a standard, was detected even in our lentisk extracts. In addition, this compound had a better α-glucosidase-inhibitory activity (81.7%) compared to the other standards tested. Therefore, it is quite coherent that the metabolites derived from epigallocatechin (M1, M3, M4 and M5) contribute to the inhibition of α-glucosidase. Previous work has established statistical models of the relationship between the phytochemical profiles obtained via LC-MS/MS and the inhibition of α-glucosidase by plant extracts [66,67,69]. The results of this work revealed that α-glucosidase-inhibitory activity was probably due to the presence of terpenoids, fatty acids, phenolic, flavonoids (epigallocatechin-3-O-gallate) and flavonoid glycosides (quercetin 3-O-glucoside) [65,67]. (-)-Epigallocatechin-3-O-gallate offers a promising hypoglycemic effect [65,70].

Most of these metabolites, especially epigallocatechin derivatives, have been reported for their ability to inhibit α-glucosidase, isolated from other plants [24,26]. Flavanol compounds (i.e., epigallocatechin, epigallocatechin gallate isomers, asinensin A, aflavin-3-gallate isomers, gallocatechin-(4α→8)-epigallocatechin and herbacetin) identified in *P. speciosa* most likely have a major role in α-glucosidase-inhibitory activity [68]. Previous studies determined several key structural features needed for monomeric flavonoids to inhibit α-glucosidase activity [71,72,73]. The compounds derived from epigallocatechin (i.e., epigallocatechin-(4α→8)-epigallocatechin (two isomers) and(epi)gallocatechin-3′-O-galloyl-(epi)gallocatechin (two isomers)) and identified in *P. lentiscus* have been described as having a major role in α-glucosidase-inhibitory activity because of the structural configuration of the double bonds conjugated to the 4-oxo and hydroxyl groups [73]. In addition, the presence of an esterified gallate group at the 3 position of the C ring has been suggested to be critical for the interaction of flavan-3-ols with the α-glucosidase enzyme [71,72]; this is the case for the molecule (epi)gallocatechin-3′-O-galloyl-(epi)gallocatechin. In addition, the substitution of sugar units or a hydroxyl group by a galloyl increases the efficacy of flavonoids against α-glucosidase [74]. The molecule 3,5-O-digalloylquinic acid probably has an important role in the inhibition of α-glucosidase. An ethnomedicinal study carried out on the hypoglycemic effects of medicinal plants showed that the leaves and fruits of *Arbutus unedo*, an antidiabetic medicinal plant, are rich in galloyl quinic acid [18]. A study by Quaresma et al.(2020) [75] reported that dihydroxy pentoside acid is among the phenolic compounds that can influence the inhibition of the active site of enzymes [75]. In this study, the butanol fraction of *Banisteriopsis argyrophylla* leaves rich in dihydroxy pentoside acid showed good inhibitory activity of α-glucosidase.

Examining the literature, in terms of lentisk phytochemistry composition, the leaf is the most studied. Shikimic acid [43], myricetin and its derivatives, including myricetin galactosides, as well as quercetin and its derivatives, were detected [64]. Luteolin-7-O-glycuronide, luteolin-3-O-rutinoside and kaempferol-3-O-di-hexose-O-pentose [19] were identified in the methanolic extracts. Other researchers have reported the presence of pentoside derivatives including quercetin-O-galloylpentoside in hydromethanolic leaf extracts [43]. Phenolic acids including gallic acid, 3-galloyl quinic acid, 5-galloyl quinic acid, 3,5-di-O-galloyl quinic acid and 1,5-di-O-galloyl quinic acid were detected in the aqueous extracts of leaves and fruits [30]. A digalloylquinic acid derivative was detected in the lentisk fruit and stem [22]. Catechin was the most representative flavanol in *P. lentiscus* leaves [23,76], while procyanidin B1 was the most abundant flavanol in the fruit extracts [77].The study conducted by Aissat et al. (2022) [78] resulted in the identification of seven flavanols, such as procyanidin B1 and B3, epigallocatechin gallate and gallocatechin and epicatechin gallate, in five different physiological stages of lentisk fruit from Algeria. There is only one study describing the phytochemical profile of the whole stem [22]. Therefore, our study is the first report on the phytochemical composition of lentisk stem barks.

Studies have shown that a plant’s geographical location has a significant influence on the composition and content of secondary metabolites [48,68]. According to the boxplots presented in Figure 7, the intensity of the identified metabolites was relatively stable (letters a, b, ab) according to the geographical origin (i.e., littoral and mountain). The slight variations were due to the many climatic and abiotic factors which can affect the biosynthesis of secondary metabolites in plants [68,79]. Colder temperatures at higher altitudes (mountains) can cause overproduction of phytochemicals [79]. Previous research has suggested that increasing altitude and consequent changes in solar radiation and temperature in plant habitats (as in Appendix A) may be strongly correlated with the contents of secondary metabolites, especially phenolics, due to their defensive function against oxidative damage [79]. On the other hand, the variation in the contents of metabolites (α-glucosidase inhibitors) depending on the part of the *P. lentiscus* plant used was more significant. The plant part plays a key role in the variation of the chemical composition of the plant extract [22]. The stem barks were characterized by an abundance of these α-glucosidase inhibitors compared to the other lentisk organs (i.e., leaves and fruits) at both sites (i.e., littoral and mountain). The results of this study suggest that lentisk stem bark from the littoral area is a good source of α-glucosidase inhibitors.

## 5. Conclusions

The results of this work revealed that the stem bark and leaves from *P. lentiscus* harvested from different areas of Algeria presented a significant activity of α-glucosidase inhibition. The stem barks harvested at the littoral had the highest inhibitory activity toward α-glucosidase. This activity was comparable to that of acarbose. A multiblock OPLS analysis was used to define the molecules that exhibited α-glucosidase inhibitory effects. The UHPLC-ESI-HRMS analysis enabled the identification of six metabolites: epigallocatchin(4a->8)-epigallocatechin (two isomers), (epi)gallocatechin-3′-O-galloyl-(epi)gallocatechin (two isomers), 3,5-O-digalloylquinic acid and dihydroxy benzoic acid pentoside. The intensities of these metabolites varieddepending on the lentisk organ and habitat. The results of our study justify scientifically and consolidate the ethnopharmacological reports on the traditional use of lentisk leaves for the treatment of type 2 diabetes. According to these results, the use of stem barks could be better adapted to the treatment of type 2 diabetes by traditional means—for example, the elaboration of traditionally improved drugs (TID) in developing countries. The antidiabetic potential of this plant could also be exploited in the field of food supplements.

Further studies on the identification of other active metabolites and their synergistic effects on α-glucosidase-inhibitory activity are still needed.

It is also important to consider preclinical and clinical trials of these lentisk extracts to determine their mechanisms of inhibition, their pharmacokinetics and their toxicology. Our results on the antidiabetic activity carried out in vitro show that the variation was quite stable according to the habitat. The same was true for the intensity variation of the metabolites that contributed most to this activity. Therefore, it would be interesting to extend the sampling to several geographical sites to better evaluate the antidiabetic activity and phytochemistry of lentisk as they relate to the environmental conditions of its habitat.

## Figures and Tables

**Figure 1 metabolites-13-00275-f001:**
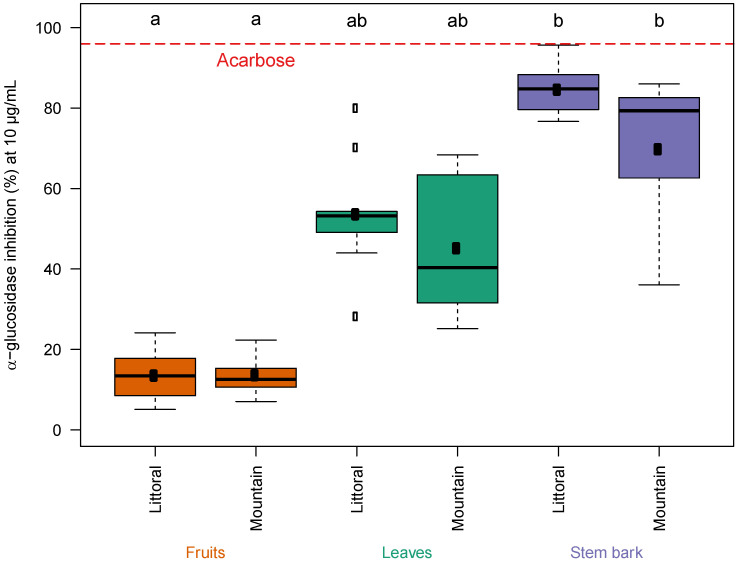
Boxplot showing α-glucosidase-inhibitory effect of lentisk leaves, stems barks and fruits from the mountain and littoral. (The means and medians are presented as black dots in the center of each box plot and by bars, respectively. The significance of the α-glucosidase inhibition variation is represented by letters (groups, a, b and ab) according to pairwise comparison through the Kruskal–Wallis test adjusted with Bonferroni correction).

**Figure 2 metabolites-13-00275-f002:**
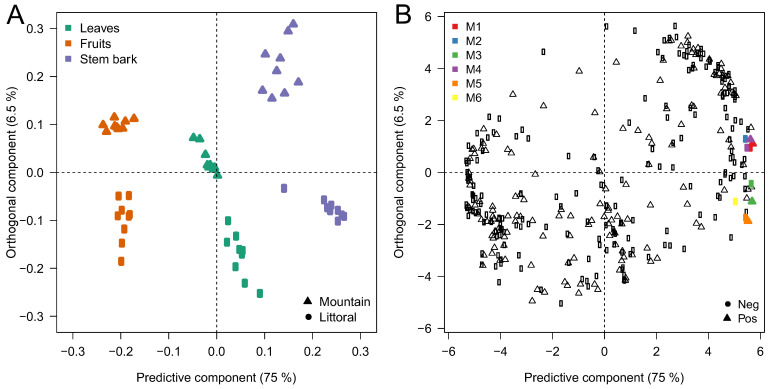
Multiblock OPLS analysis performed on the UHPLC-ESI-HRMS and α-glucosidase inhibition data: (**A**) score plot—lentisk samples: ▪ Littoral/Δ Mountain; (**B**) loading plot—metabolites detected: ▪ ions with negative ionization/Δ ions with positive ionization: metabolites; M1: epigallocatechin(4a->8)epigallocatechin (1); M2: 3,5-O-digalloylquinic acid; M3: (epi)gallocatechin-3′-O-galloyl-(epi)gallocatechin (1); M4: epigallocatechin(4a->8)epigallocatechin (2); M5: (epi)gallocatechin-3′-O-galloyl-(epi)gallocatechin (2); M6:dihydroxy benzoic acid pentoside.

**Figure 3 metabolites-13-00275-f003:**
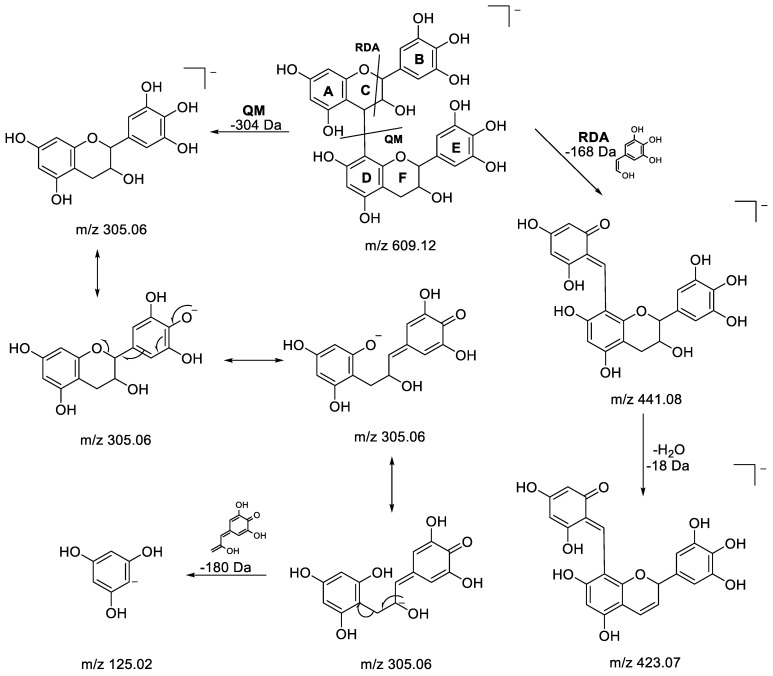
Proposed fragmentation pathways for epigallocatechin(4a->8)epigallocatechin.

**Figure 4 metabolites-13-00275-f004:**
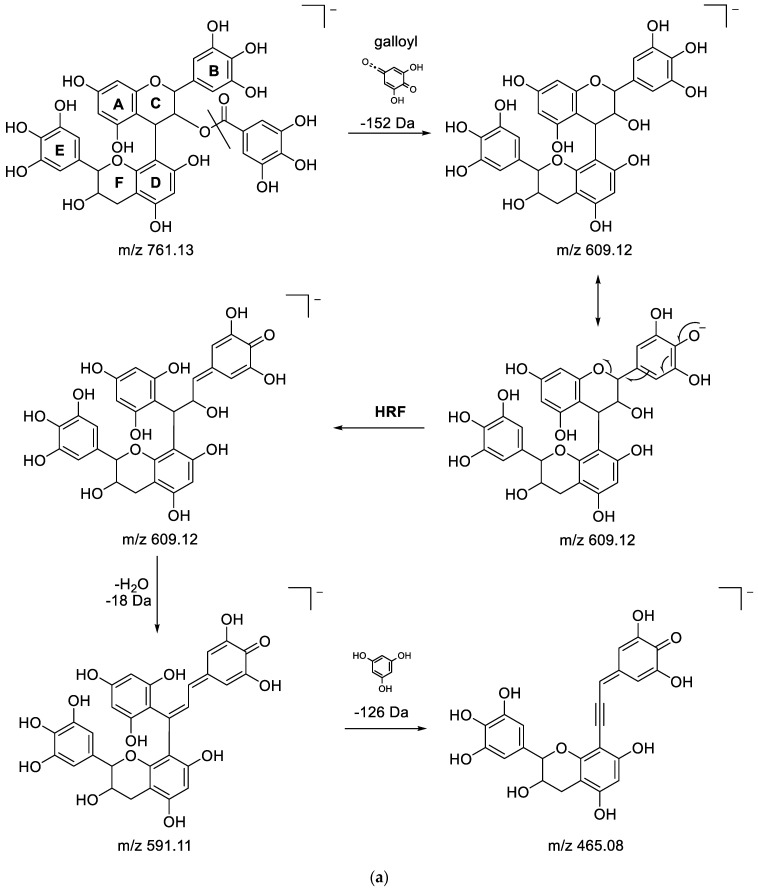
(**a**) Fragmentation pathways of (epi)gallocatechin-3′-O-galloyl-(epi)gallocatechin showing the formation of the fragmentations at *m/z* 609.12,591.11 and 465.08. (**b**) Fragmentation pathways of (epi)gallocatechin-3′-O-galloyl-(epi)gallocatechin showing the formation of the fragment ions at *m/z* 423.08 and 305.06.

**Figure 5 metabolites-13-00275-f005:**
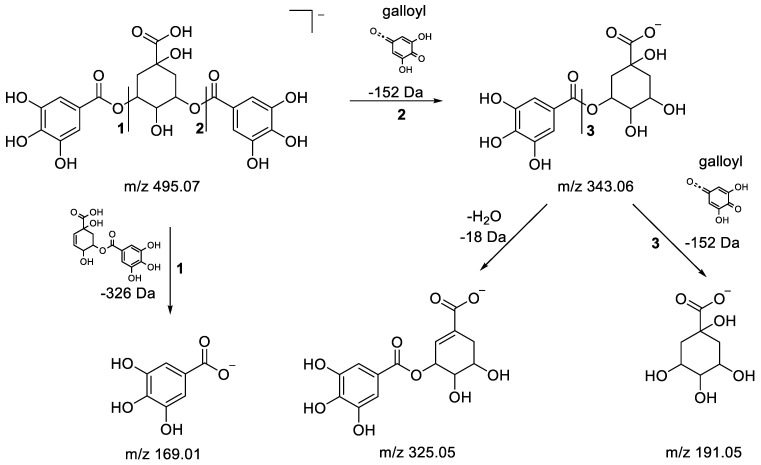
Fragmentation pathway of 3,5-O-digalloylquinic acid.

**Figure 6 metabolites-13-00275-f006:**
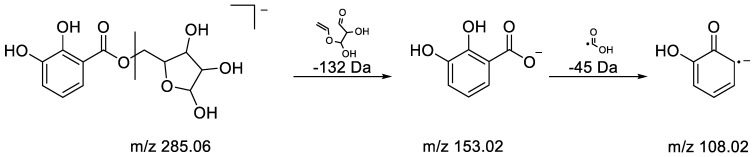
Fragmentation pathways of dihydroxy benzoic acid pentoside.

**Figure 7 metabolites-13-00275-f007:**
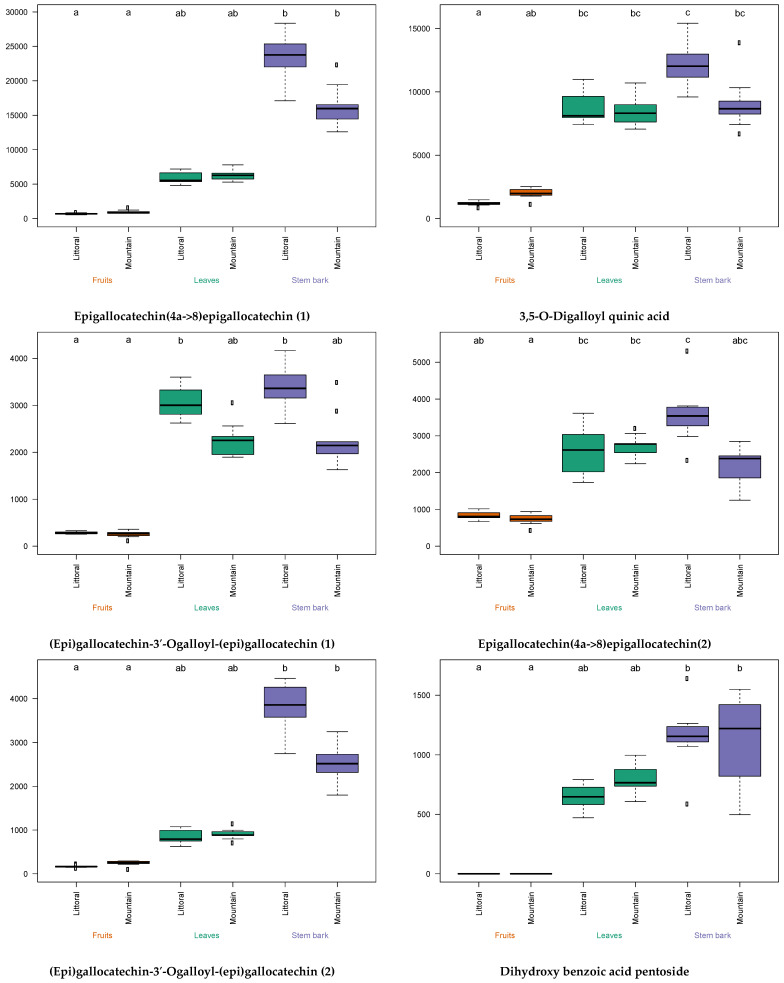
Box plots of the metabolite contents in different organs—leaves, stem barks and fruits—from the mountain and littoral. (The significance of the metabolite intensity variation is presented by letters (groups, a, b, c, ab, bc and abc) according to pairwise comparison through the Kruskal-Wallis test adjusted with Bonferroni correction).

**Table 1 metabolites-13-00275-t001:** Information on lentisk samples harvested on the mountain and on the littoral.

Sites	Collected Date	54 Samples (9 Trees × 3 Organs × 2 Sites)	Morphological Variations	Longitude	Latitude	Altitude	Annual Temperature	
Site 1 Ait-Irane Mountain	Mid-October 2019	9 lentisk trees (3 organs: leaves, stem bark and fruit) = 27 samples	Unremarkable for the same organ type	36°29′58.3″ N	4°04′43.4″ E	876 m	Min	Max
							0.5 °C	31.9 °C
Site 2 Tigzirt Littoral	Mid-October 2019	9 lentisk trees (3 organs: leaves, stem bark and fruit) = 27 samples	Unremarkable for the same organ type	36°53′43.0″ N	4°11′00.4″ E	13 m	Min	Max
							7.4 °C	27.9 °C

No site-specific morphological variations for the same organ type. More information on the climatic conditions at these two sites is presented in Appendix A.

**Table 2 metabolites-13-00275-t002:** Phenolic compounds identified in *P. lentiscus* extracts by UHPLC-ESI-HRMS in the negative mode. (The equivalent information for the positive mode is presented in Appendix A).

N° Metabolite	RT (min)	*m/z* (obs) [M-H]^−^	Error (ppm)	HDMS^E^ Fragment Ions (Intensity, %)	Molecular Formula	Suggested Compound	Ref
1	0.45	609.1242	−1.26	441.0825 (2.99), 423.0722 (100), 305.0667 (29.49), 261.0397 (15.71), 219.0659 (14.46), 177.0186 (26.94), 125.0241(18.09)	C_30_H_26_O_14_	Epigallocatechin(4a>8)epigallocatechin	[55,56,57]
2	0.55	495.0774	−2.03	343.0661(100), 325.0550(15.74), 191.0551(87.47), 169.0136(37.71)	C_21_H_20_O_14_	3,5-O-Digalloylquinic acid	[43]
3	0.56	761.1339	−3.46	609.1228(12.42), 591.1134(5.98), 465.0820 (12.76), 423.0820(100), 305.0658(46.73)	C_37_H_30_O_18_	(Epi)gallocatechin-3′-Ogalloyl-(epi)gallocatechin	[58]
4	0.61	609.1233	−1.51	423.0716 (100)	C_30_H_26_O_14_	Epigallocatechin(4a->8)epigallocatechi-n (isomer)	[57,58,59]
5	0.72	761.1344	−2.28	423.0715 (100)	C_37_H_30_O_18_	(Epi)gallocatechin-3′-Ogalloyl-(epi)gallocatechin (isomer)	[58]
6	0.73	285.0608	−2.76	108.0209 (100)	C_12_H_14_O_8_	Dihydroxy benzoic acid pentoside	[60]

## Data Availability

Data are contained within the article or the Appendix A.

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
