# Peer review of "Metabolomics-Based Profiling via a Chemometric Approach to Investigate the Antidiabetic Property of Different Parts and Origins of Pistacia lentiscus L."

_metabolites, 2023, doi:10.3390/metabo13020275_

Round 1
Reviewer 1 Report
In this paper, the authors uesd a metabolic Profiling approach to study the antidiabetic property of Pistacia Lentiscu. The work is a significant contribution to the field and is well-organized and well-written. However, a couple of comments should be addressed.
The introduction needs to be improved by clearly showing what knowledge gaps are identified from literature review. And how the authors connect the knowledge gaps to their research goals. Please reason the relevance of your research goals, as has not been sufficiently highlighted in the current version.
The methods can be improved. The authors claim that they used a metabolomics approach. However, metabolite datasets are not available. I strongly recommend the authors deposit their metabolomics data to a data repository and include the accession numbers in the Methods.
Figures should be appropriately described and labeled. The shape of the markers in Figure 1 should be desribed in figure legend.
The discussion part for this manuscript can be improved. The discussion section is the main part of a paper. It should not just summarize the key results of the study, but to highlight the insights and the applicability of your results for future work. Also, please always avoid using the words "first time".
Author Response
Reviewer 1
Comment 1:
The introduction needs to be improved by clearly showing what knowledge gaps are identified from literature review. And how the authors connect the knowledge gaps to their research goals. Please reason the relevance of your research goals, as has not been sufficiently highlighted in the current version.
Reply 1:
The introduction is enhanced by highlighting:
(i) The literature review and identified knowledge gaps (line 74-80) are now highlighted as follows: “Even though there are many ethnobotanical pharmaceutical uses of P. lentiscus [29], very few studies [23,30] have been carried out to explore the possibilities of using this species as an antidiabetic agent. These studies were carried out by Foddai et al. [30] on aqueous leaf and fruits extract and Mehenni et al. [23] on ethanolic extracts. Most of the research on P. lentiscus has been conducted to find out its antioxidant [31,32,33,34,35,36,37], antimicrobial [38,39,40,41], anticancer [42,43,44,45] and anti-inflammatory potential [46,47,48,49].”
- ii) The link between these gaps and the objectives of this article (line 80-92) is now written as: “So far, no study has reported a metabolomic analysis to identify inhibitors of antidiabetic potential in different lentisk organs. It is known that different geographical regions can affect the composition and biological activity of plants [50]. Therefore, our study is the first work that aimed to achieve the following:
- Evaluate α-glucosidase inhibition in lentiscus samples and identify compounds more positively correlated to α-glucosidase inhibition using a metabolomics approach;
- Study the variation of this inhibition according to the organ (i.e., leaf, stem bark and fruit) and the geographical origin (i.e., mountain and littoral) of the plant lentiscus. This metabolomic method was based on the link between phytochemical profiling by ultra-high-performance liquid chromatography coupled to high-resolution mass spectrometry (UHPLC-ESI-HRMS) and α-glucosidase dosage data. “
iii) The relevance of these objectives (line 93-96) is now: “This study could contribute to the valorization of the antidiabetic potential of this resource in the pharmaceutical field, for example the elaboration of traditionally improved drugs (TID) in developing countries. The anti-diabetic potential of this plant could also be exploited in the field of food supplements. “
Comment 2:
The methods can be improved. The authors claim that they used a metabolomics approach. However, metabolite datasets are not available. I strongly recommend the authors deposit their metabolomics data to a data repository and include the accession numbers in the Methods.
Reply 2:
The explanation of the metabolomic approach has been improved (please see line 191-205 in the revised version) and the metabolomic data are deposited (as new supplementary Table S3 and S4), their accession number is included in the methods (please see line 193 in the revised version; please see table S3 and S4):
Supplementary Materials: The following supporting information can be downloaded at: www.mdpi.com/xxx/s1, Figure S1: Boxplot showing the α-glucosidase inhibitory effect of lentisk leaves, stems barks and fruits from the mountain and littoral; Figure S2: Contribution of the two blocks (neg and pos) to the constitution of the OPLS model; Table S1. Weather data collected between 1999 and 2019 at Ait Irane, Algeria; Table S2. Weather data collected between 1999 and 2019 at Tigzirt, Algeria. Table S3: Metabolomic data (OPLS, matrix-negative ionization); Table S4: Metabolomic data (OPLS, matrix-positive ionization); Table S5: UHPLC-ESI-HRMS data results and OPLS coordinates; Table S6: Phenolic compounds identified in P. lentiscus extracts by UHPLC-ESI-HRMS in the positive mode.
Comment 3:
Figures should be appropriately described and labeled. The shape of the markers in Figure 1 should be described in figure legend.
Reply 3:
The location, shape of the markers in Figure 1 have been described in the figure legend: (A) score plot—lentisk samples: â–ª Littoral Δ Mountain; (B) loading plot—metabolites detected: â–ª ions with negative ionization Δ ions with positive ionization, (line 304-305)
Comment 4
The discussion part for this manuscript can be improved. The discussion section is the main part of a paper. It should not just summarize the key results of the study, but to highlight the insights and the applicability of your results for future work. Also, please always avoid using the words "first time".
Reply 4:
The prospects and applicability of the results of this research for future work have been highlighted (line 597-601) as: for example the elaboration of traditionally improved drugs (TID) in underdeveloped countries. The anti-diabetic potential of this plant could also be exploited in the field of food supplements. Further studies on identification of other active metabolites and their synergistic effect against α-glucosidase inhibitory activity are still needed. “
Also, the words "first time" are deleted in the text of this revised manuscript
Reviewer 2 Report
Comment and suggestion for authors:
Manuscript ID: metabolites-2137098
Type: Article
Titled: Metabolomics-based profiling via a chemometric approach to investigate the antidiabetic property of different parts and origins of Pistacia Lentiscus
1) In the title, the specific epithet of this species name must be written using lowercase and the author who names this plant should be added to this species name, so “Pistacia Lentiscus” in the title should be changed to be “Pistacia lentiscus L.”.
2) “L.” is the name of the author who names this plant and MUST NOT BE WRITTEN IN ITALIC (only the species name/scientific name have to be written in italic.). Please, check and correct this issue in the whole manuscript.
3) Please check the whole manuscript, after using the full scientific name “Pistacia lentiscus L.” in Line 18, the others have to use abbreviation of genus name “P. lentiscus”.
4) Line 89-91, the authors mentioned that “In the present work, the different aerial parts (i.e., leaves, stem barks and fruits) of the plant Pistacia lentiscus were collected from two sites (i.e., mountain and littoral) in Tizi-Ouzou (Algeria). The 54 samples obtained were characterized phytochemically and biologically.”. The table to provide the information of the 54 samples such as collected site, collected date, GPS of each collection site, and the morphological variation in the characters of each sample should be provided and clearly discuss with the results.
5) The figure 2, “B” is overwritten the description of X-axis. Please, check and edit this figure.
6) The greater number of another related published works should be added to discuss with the results from this current study, especially on “α-Glucosidase Inhibitory Activity”, “UHPLC-ESI-HRMSidentification of α-glucosidase inhibitory metabolites” and “Fragmentation pathways of (epi)gallocatechin-3’-O-galloyl-(epi)gallocatechin”.
7) There are some spelling mistakes and grammatical error found in this manuscript. Please, check and correct.
Author Response
Reviewer 2
Comment 1:
In the title, the specific epithet of this species name must be written using lowercase and the author who names this plant should be added to this species name, so “Pistacia Lentiscus” in the title should be changed to be “Pistacia lentiscus L.”
Reply 1:
The species name "Pistacia Lentiscus" in the title was changed to "Pistacia lentiscus L.".
Comment 2:
“L.” is the name of the author who names this plant and MUST NOT BE WRITTEN IN ITALIC (only the species name/scientific name have to be written in italic.). Please, check and correct this issue in the whole manuscript.
Reply 2:
The italicization of the letter "L" the name of the author who names this plant has been cancelled throughout the manuscript.
Comment 3:
Please check the whole manuscript, after using the full scientific name “Pistacia lentiscus L.” in Line 18, the others have to use abbreviation of genus name “P. lentiscus”.
Reply 3:
The abbreviation of the genus name "P. lentiscus" has been applied throughout this article.
Comment 4:
Line 89-91, the authors mentioned that “In the present work, the different aerial parts (i.e., leaves, stem barks and fruits) of the plant Pistacia lentiscus were collected from two sites (i.e., mountain and littoral) in Tizi-Ouzou (Algeria). The 54 samples obtained were characterized phytochemically and biologically.”. The table to provide the information of the 54 samples such as collected site, collected date, GPS of each collection site, and the morphological variation in the characters of each sample should be provided and clearly discuss with the results.
Reply 4:
- In material and methods: in the plant collection section (line 109-124); the date of sampling (line 124), the GPS of each sampling site (line 118/121) have been added
- The tables (S1 and S2: line 119/122) covering the climatic conditions of each sampling site is provided in the supplementary data
- The climatic condition sampling informations presented in plant collection section and tables are clearly discussed with the results (line 569-576): These slight variations were due to the many climatic or abiotic factors, which can affect the biosynthesis of secondary metabolites in plants [72,83]. Colder temperatures at higher altitudes (mountains) can cause overproduction of phytochemicals [83]. Previous research has suggested that increasing altitude and consequent changes in solar radiation and temperature in plant habitats (as in Table S1 et S2) may be strongly correlated with the content of secondary metabolites, especially phenolics, due to their defensive function against oxidative damage [83]
Comment 5:
The figure 2, “B” is overwritten the description of X-axis. Please, check and edit this figure.
Reply 5:
Figure 2, "B", has been modified.
Comment 6:
The greater number of another related published works should be added to discuss with the results from this current study, especially on “α-Glucosidase Inhibitory Activity”, “UHPLCESI-HRMS identification of α-glucosidase inhibitory metabolites” and “Fragmentation pathways of (epi)gallocatechin-3’-O-galloyl- (epi)gallocatechin”.
Reply 6:
Other previous works on "Inhibitory activity of α-Glucosidase", "UHPLC-ESI-HRMS identification of α-glucosidase inhibitory metabolites" and "Fragmentation pathways of (epi)gallocatechin-3'-O-galloyl-(epi)gallocatechin" have been added to better discuss the results in the discussion part (please see line 439-497/ line 518-524)
Line 493-497: The activity of α-glucosidase inhibition is tested on several other plants [69,70,71]. Plant extracts including Tamarix nilotica (IC50=33.3 µg/ml) [69], Tetracera scandens (IC50=3.4 µg/ml) [70], Cosmos caudatus (IC50=39.1 µg/ml) [71] showed promising α-glucosidase inhibition. These activities are comparable to those of acarbose and the results found in the present study
Line 518-524: Previous work has established statistical models between the pytochemical profile via LC-MS/MS and the inhbition of α-glucosidase from plants extracts [70,71,73]. The results of this work revealed that α-glucosidase inhibitory activity was probably due to the presence of terpenoids, fatty acids, phenolic , flavonoids (epigallocatechin-3-O-gallate) and flavonoid glycosides (quercetin 3-O-glucoside) [69,71]. (-)-Epigallocatechin-3-O-gallate offers promising hypoglycemic effect [69,74]
Line 375-388: Fragmentation pathways of (epi)gallocatechin-3'-O-galloyl-(epi)gallocatechin" have been illustrated (figure 4a, 4b) and compared to literature.
Comment 7:
There are some spelling mistakes and grammatical error found in this manuscript. Please, check and correct.
Reply 7:
Spelling and grammatical errors found in this manuscript have been checked and corrected
Line 228: “The inhibitory effect on α-glucosidase was evaluated by calculating the percentage of inhibition of this enzyme in a 96-well plate” is replaced by “the inhibitory effect on α-glucosidase was estimated by evaluating the percentage of inhibition of this enzyme in a 96-well plate”
Line 196: identifying the metabolites that were located at the right end is replaced by identifying metabolites located on the right end
The MDPI English editing reference is: “English-Editing-Certificate-54629”.
Reviewer 3 Report
The manuscript titled “Metabolomics-Based Profiling via A Chemometric Approach to 2 Investigate the Antidiabetic Property of Different Parts and 3 Origins of Pistacia Lentiscus” is well written.
Nevertheless, some issues here:
1. The way the matrix was developed need to be disclosed further. Is this based on the chromatogram, if so, how the binning was performed.
2. Add more discussion on micro-climate and secondary metabolites production. This to avoid generalisation of using stembark over the others across the globe.
3. Add literature study on the epigallocatechin derivatives from other species with similar activity.
Author Response
Reviewer 3
Comment 1:
The way the matrix was developed need to be disclosed further. Is this based on the chromatogram, if so, how the binning was performed?
Reply 1:
The way the matrix was developed has been explained in more detail (please see line 191-204 and 270-276 in the revised version). it is based on the chromatogram (line 203), the way of binning has been developed in more detail (please see line 191-204 and 270-276)
Comment 2
Add more discussion on micro-climate and secondary metabolites production. This to avoid generalization of using stembark over the others across the globe.
Reply 2:
The discussion on microclimate and secondary metabolite production has been expanded (line 566-576):
Line 566-576: Studies have shown that the geographical location has a significant influence on the composition and content of secondary metabolites [30,49].According to the boxplots presented in Figure 7, the intensity of the identified metabolites was relatively stable (letters a, b, ab) according to the geographical origin (i.e., coastal and mountain).These slight variations were due to the many climatic or abiotic factors, which can affect the biosynthesis of secondary metabolites in plants [72,83]. Colder temperatures at higher altitudes (mountains) can cause overproduction of phytochemicals [83]. Previous research has suggested that increasing altitude and consequent changes in solar radiation and temperature in plant habitats (as in Table S1 et S2) may be strongly correlated with the content of secondary metabolites, especially phenolics, due to their defensive function against oxidative damage [83]
Comment 3
Add literature study on the epigallocatechin derivatives from other species with similar activity.
Reply 3:
The literature review on epigallocatechin derivatives from other species with similar activity has been inserted in the discussion section (please see line 518-540).
Round 2
Reviewer 2 Report
According to the author reply to my Comment 4,
"
Reply 4:
- In material and methods: in the plant collection section (line 109-124); the date of sampling (line 124), the GPS of each sampling site (line 118/121) have been added
- The tables (S1 and S2: line 119/122) covering the climatic conditions of each sampling site is provided in the supplementary data
- The climatic condition sampling informations presented in plant collection section and tables are clearly discussed with the results (line 569-576): These slight variations were due to the many climatic or abiotic factors, which can affect the biosynthesis of secondary metabolites in plants [72,83]. Colder temperatures at higher altitudes (mountains) can cause overproduction of phytochemicals [83]. Previous research has suggested that increasing altitude and consequent changes in solar radiation and temperature in plant habitats (as in Table S1 et S2) may be strongly correlated with the content of secondary metabolites, especially phenolics, due to their defensive function against oxidative damage [83]"
I strongly recommend the authors to put this information as the additional table to the main manuscript to provide the information of the 54 samples such as collected site, collected date, GPS of each collection site, and the morphological variation in the characters of each sample should be provided and clearly discuss with the results and make it easy to understand for the readers.
Author Response
Dear Reviewer 2,
As demanded in Comment 4:
I strongly recommend the authors to put this information as the additional table to the main manuscript to provide the information of the 54 samples such as collected site, collected date, GPS of each collection site, and the morphological variation in the characters of each sample should be provided and clearly discuss with the results and make it easy to understand for the readers.
Reply 4:
- Line 116-120: For this study, we selected two geographical sites in the Tizi Ouzou region: Ait-Irane, on the mountain and Tigzirt on the littoral. 54 samples were collected. The information of the 54 samples such as collected site, collected date, GPS of each collection site, and the morphological variation in the characters of each sample was provided in Table 1
Please see: Table 1. Information on lentisk samples harvested on the mountain and on the littoral: line 127
No site-specific morphological variations for the same organ type
More information on the climatic conditions at these two sites is presented in Tables S1 and Table S2
- The climatic condition sampling informations presented in plant collection section and tables are clearly discussed with the results (line 624-631): These slight variations were due to the many climatic or abiotic factors, which can affect the biosynthesis of secondary metabolites in plants [72,83]. Colder temperatures at higher altitudes (mountains) can cause overproduction of phytochemicals [83]. Previous research has suggested that increasing altitude and consequent changes in solar radiation and temperature in plant habitats (as in Table S1 et S2) may be strongly correlated with the content of secondary metabolites, especially phenolics, due to their defensive function against oxidative damage [83]"

Round 3
Reviewer 2 Report
Dear Authors,
Thanks for the revised version. There is only one last comment for your manuscript. The altitude of the 2 sites (876 m VS 13 m) in this study is quite different, is there any difference between the plant samples from these 2 sites? Why? Please, described this information in your manuscript.
Sincerely,
Reviewer
Author Response
Dear Reviewer2,
Please find the reply to the last comment of the reviewers for article references MDPI Metabolites 2137098.
Best regards
Eric Gontier and Chabha Sehaki (for the authors)
Comment:
There is only one last comment for your manuscript. The altitude of the 2 sites (876 mVS 13 m) in this study is quite different, is there any difference between the plant samples from these 2 sites? Why? Please, described this information in your manuscript.
Reply:
Line 117-123: 54 samples were collected. The informations of the 54 samples such as collected site, collected date, GPS of each collection site, and the morphological variation in the characters of each sample were provided in Table 1. Now, we added the further comment: “The lentisk samples harvested on the two sites (mountain and littoral) do not show remarkable morphological differences according to the difference in altitude of the harvesting site” in the text before Table 1.